# Identification and categorisation of relevant outcomes for symptomatic uncomplicated gallstone disease: in-depth analysis to inform the development of a core outcome set

Moira Cruickshank,[1] Rumana Newlands,[1] Jane Blazeby [ID],[2] Irfan Ahmed,[3] Mohamed Bekheit,[3,4] Miriam Brazzelli [ID],[1] Bernard Croal [ID],[5] Karen Innes [ID],[1] Craig Ramsay [ID],[1] Katie Gillies [ID] [1]

► Prepublication history and additional online supplemental material for this paper are available online. To view these files, please visit the journal online. To view these files, please visit the journal online (http://dx.doi.org/10.1136/bmjopen-2020-045568).

MC and RN are joint first authors.

For numbered affiliations see end of article.

**Correspondence to**
Dr Katie Gillies;
k.gillies@abdn.ac.uk

## ABSTRACT

**Background** Many completed trials of interventions for uncomplicated gallstone disease are not as helpful as they could be due to lack of standardisation across studies, outcome definition, collection and reporting. This heterogeneity of outcomes across studies hampers useful synthesis of primary studies and ultimately negatively impacts on decision making by all stakeholders. Core outcome sets offer a potential solution to this problem of heterogeneity and concerns over whether the 'right' outcomes are being measured. One of the first steps in core outcome set generation is to identify the range of outcomes reported (in the literature or by patients directly) that are considered important.

**Objectives** To develop a systematic map that examines the variation in outcome reporting of interventions for uncomplicated symptomatic gallstone disease, and to identify other outcomes of importance to patients with gallstones not previously measured or reported in interventional studies.

**Results** The literature search identified 794 potentially relevant titles and abstracts of which 137 were deemed eligible for inclusion. A total of 129 randomised controlled trials, 4 gallstone disease specific patient-reported outcome measures (PROMs) and 8 qualitative studies were included. This was supplemented with data from 6 individual interviews, 1 focus group (n=5 participants) and analysis of 20 consultations. A total of 386 individual recorded outcomes were identified across the combined evidence: 330 outcomes (which were reported 1147 times) from trials evaluating interventions, 22 outcomes from PROMs, 17 outcomes from existing qualitative studies and 17 outcomes from primary qualitative research. Areas of overlap between the evidence sources existed but also the primary research contributed new, unreported in this context, outcomes.

**Conclusions** This study took a rigorous approach to catalogue and map the outcomes of importance in gallstone disease to enhance the development of the COS 'long' list. A COS for uncomplicated gallstone disease that considers the views of all relevant stakeholders is needed.

### Strengths and limitations of this study

► This outcome map review is the first to describe the heterogeneity in outcome reporting within uncomplicated symptomatic gallstone disease.
► There is a detailed analysis of all reported outcomes from a range of study designs (both primary and secondary) reporting outcomes of clinical and/or patient relevance.
► A mixed-methods approach was used in both collection and analysis of data.
► Only studies reported in English language were included in the analysis
► Quality assessment of included studies was not conducted as the main purpose of the review was to extract clinical outcomes not to assess intervention effectiveness.

## BACKGROUND

Gallstone disease (cholelithiasis) is one of the most common gastrointestinal disorders worldwide. The prevalence of gallstones is approximately 10%–15% in adult populations and they are more common in women and people over the age of 40.[1] Approximately 80% of those affected by gallstone disease are asymptomatic and can remain so for many years without requiring treatment. However, around 20% of patients with gallstones become symptomatic and develop gallstone-related complications. These patients are offered symptom control and/or surgical/endoscopic intervention.[2] A significant number of these patients remain symptomatic only (ie, experiencing pain) without developing gallstone-related complications.[2]

Recommendations from the recent National Institute for Health and Care Excellence guideline on Gallstone Disease

have clearly demonstrated insufficient information for patients with gallstones on the effect of cholecystectomy on patient outcomes.[3] The guideline recommends 'research is needed to establish the long-term patient benefits and harms, so that appropriate information can be provided to patients to aid decision-making and long-term management of their condition.'[3] However, many completed trials are not as helpful as they could be due to lack of standardisation across studies, outcome definition, collection and reporting. This heterogeneity of outcomes across studies also hampers useful synthesis of primary studies in meta-analyses and ultimately negatively impacts on decision making by all stakeholders. In addition to the heterogeneity of outcomes currently reported and the problems this causes, measuring the wrong outcomes (ie, those that are not valued by clinicians or, more importantly, patients) could also be a real risk for many studies if stakeholders are not consulted during the trial design process. One way that these problems with heterogeneity and relevance to stakeholders can be addressed is through the development and use of core outcome sets.[4 5] There is currently no agreed published core outcome set for evaluating interventions to treat symptomatic uncomplicated gallstone disease.

Core outcome sets (COS) aim to define a minimum set of outcomes that should be considered essential for the evaluation and reporting of specific interventions or conditions (ie, the set of outcomes that should always be considered and ideally measured in any evaluation).[4 5] There is a growing body of literature to provide support for development of core outcome sets.[5] Specifically, they are developed using consensus methods involving stakeholder groups, such as health professionals and patients, so as to ensure that the outcomes being defined are both clinically and personally relevant for the individuals involved.[4 5] Assessment of a core outcome set is not expected to be mutually exclusive to the measurement of other outcomes. However, a core set will foster greater consistency in outcome reporting between studies and lead to more meaningful data being available to contribute to meta-analysis.[4 5] Moreover, core outcome sets can minimise the threat of outcome reporting bias by ensuring consistency between what is measured and what is reported.[4 5] Ultimately, they should improve the overall efficiency and quality of the evidence on which healthcare decisions can be made.

A core outcome set for uncomplicated gallstone disease is currently being developed. Details of this project have been registered and included in the Core Outcome Measures in Effectiveness Trials (COMET) Initiative database.[6] Outcome mapping is an important step in the development of core outcome sets, to present and catalogue the outcomes reported to date, and links the literature review and the subsequent process of consultation and consensus.[7] Therefore, the objectives of this paper are to document the outcome mapping process in the development of the core outcome set for symptomatic uncomplicated gallstone disease.

## METHODS

The protocol for development of the COS is available on the COMET website: http://www.comet-initiative.org/studies/details/927?result=true.

### Identification of outcomes relevant for symptomatic uncomplicated gallstone disease

The identification of outcomes was informed by two sources: existing evidence and new primary research. The specifics of these are detailed below.

1. Identification of outcomes from existing literature:
   a. Outcomes reported in trials of interventions for symptomatic uncomplicated gallstone disease.
   b. Content analysis of individual items within disease-specific patient-reported outcome measures (PROMs).
   c. Outcomes from exploratory studies reported by patients with a lived experience of symptomatic uncomplicated gallstone disease.
2. Identification of outcomes of relevance to patients from new primary research.
   a. Interviews and focus groups with patients with a diagnosis of symptomatic uncomplicated gallstone disease and a range of treatments planned or received.
   b. Audiorecordings of consultations for a clinical trial evaluating treatments for symptomatic uncomplicated gallstone disease.

Methods for each of these outcome identification stages are specified below.

### Identification of outcomes from existing literature: outcomes reported in trials of interventions for symptomatic uncomplicated gallstone disease and identification of disease specific PROMs

Reported outcomes of interventions for symptomatic uncomplicated gallstone disease were identified by updating the search strategy for a recent systematic review (Brazzelli *et al*[8]; this review included two randomised controlled trial, RCTs), by conducting a search for relevant PROMs and by screening the reference lists of relevant Cochrane reviews. In addition, reference lists of systematic reviews identified by the search strategy were checked for relevant RCTs.

### Search methods for identification of studies

Extensive electronic searches were undertaken to identify trials for a project on the clinical effectiveness of cholecystectomy and these are reported in full elsewhere.[8] The databases searched included MEDLINE (1946 to week 37 2012), MEDLINE-in-process (10 September 2012), Embase (1974 to 2012 September 10), Science Citation Index (1970 to 12 September 2012), BIOSIS (1956 to 12 September 2012) and the Cochrane Central Register of Controlled Trials (Issue 9–12, 2012). Studies identified from these searches were used to elicit reported outcomes. The MEDLINE and EMBASE searches were

updated in May 2016 (September 2012–May 2016) to identify more recent relevant trials.

In addition, a specific search of MEDLINE and EMBASE was undertaken to identify studies that report PROM outcome data for cholecystitis, with records retrieved by the main search for trials excluded to avoid duplication of the results. This search was undertaken in May 2016 (1980 to May 2016). The search strategies for MEDLINE and EMBASE are reported in online supplemental appendix 1. Inclusion criteria for eligible studies were as follows; Participants: Adults aged over 18 years with symptomatic uncomplicated gallstone disease. Intervention and comparator: Any intervention (surgical or non-surgical management, ie, expectant management or dietary advice or medical therapy) used to manage symptomatic uncomplicated gallstone disease in adults. Outcomes: All reported outcomes well eligible for inclusion. Excluded studies included those focusing on asymptomatic gallstone disease or on acute severe cholecystitis, cholangitis, or pancreatitis were not considered suitable for inclusion. In addition, studies including 'complex' gallstone cases that is, empyema, ascending cholangitis and gallstone ileus, were excluded. Reports published in non-English languages for which a translation could not be organised were also excluded. In addition, lists of included and excluded studies reported in several relevant Cochrane reviews were checked by one reviewer (MC) for potentially relevant studies.[9–14]

## Study selection and data extraction

One reviewer (MC) screened all titles and abstracts identified by the two search strategies and a second reviewer (RN) checked a 10% random sample. All full-text papers considered potentially eligible were screened by one reviewer (MC) and checked by a second reviewer (RN). One reviewer (MC) extracted details of all outcomes reported (verbatim) and any reported definition of outcomes provided by the authors (eg, operating time may have been defined and reported by some studies as 'interval between initial skin incision and sin closure' others 'duration of surgery' etc. The definition reported by study authors was used to when deduplicating items into a shorter list. Data were recorded in a Microsoft Excel file. A 10% sample was checked by a second reviewer (RN). Other relevant data (ie, study and participant characteristics) were extracted by one reviewer (MBe) and checked by a second reviewer (MC). At all stages, disagreement between reviewers was resolved by discussion.

## Data extraction from PROMs

From the list of outcomes reported in trials of interventions for symptomatic uncomplicated gallstone disease described above, disease specific PROMS were identified and supplemented with the studies identified in the search. Data were extracted by one reviewer (KG) who recorded the name of the PROM(s), the reported PRO scales and individual verbatim items. The individual verbatim items from each PROM were analysed using an inductive content analysis approach and informed by previous PROM coding work.[15] All PROM items were systematically categorised into conceptual health domains according to the aspect which they aim to capture. Health domains were generated inductively from the identified individual items. Domain mapping was conducted two authors (KG and JB) independently with any conflicts resolved through discussion.

## Identification of outcomes from existing literature: outcomes reported as important by patients with a lived experience of uncomplicated gallstone disease

### Search methods for identification of studies

A search for relevant qualitative studies was undertaken in August 2016 in the Ovid versions of MEDLINE (from 1966 to 2016). The search strategies combined search terms for cholecystitis, cholecystectomy, with terms for qualitative research (online supplemental appendix 1). Inclusion criteria included (1) studies that have explored (using observations, interviews, focus groups and other qualitative methods) participants' lived experiences of gallstones with specific reference to outcomes of importance. Exclusion criteria included (1) studies that have not used qualitative methods; (2) any review articles, conference abstracts and those with no full-text articles published or non-English language articles.

### Study selection and data extraction

One researcher (RN) screened all abstracts and another (KG) screening a random 10% sample. Full-text articles were obtained for those that were potentially relevant. Two researchers (RN and KG) reviewed all potentially relevant articles to ensure they met the inclusion criteria. To identify additional relevant studies, the reference lists of the included studies were also examined.

Data on study characteristics such as author, publication date, country, focus of investigation, data collection methods, number of participants and details on sample size were extracted. Additionally, two authors (KG and RN) independently extracted data from two main sources reporting study findings: (1) Direct quotes from participants and (2) Authors interpretations of participants quotes. These data were recorded verbatim and analysed to identify 'descriptive' thematic codes. Constant comparison method was used to compare findings across studies and an inductive thematic synthesis was undertaken to generate a list of themes and subthemes (focused on outcomes) from the data to map across the presurgery and postsurgery timeline.[16 17] Throughout this process, the description and wording of the themes were continually revised, and notes made as to how themes and/or subthemes related and how some could be merged. These findings were discussed further with the research team to finalise the themes across the studies and these were considered, where appropriate, as domains relevant for inclusion in the development of the COS.

### Identification of outcomes of relevance to patients from new primary research

In addition to outcomes reported in existing literature, we conducted primary qualitative research to further inform the identification of outcomes of relevance to patients. Three activities were conducted:

1. Interviews with patients with a diagnosis of symptomatic uncomplicated gallstone disease.
2. A focus group with patients who had undergone cholecystectomy.
3. Analysis of audioconsultations from a trial comparing surgical versus medical management of symptomatic uncomplicated gallstone disease.

#### Participant identification and invitation

Potential participants for the interviews were identified from an ongoing trial comparing laprascopic cholecystectomy with observation/conservative management for preventing recurrent symptoms in adutls with uncomplicated symptomatic gallstones (CGALL trial). Participants were provided with a study participant information leaflet (PIL) either in the clinic or posted to the participant if a decision about CGALL trial entry was made later.[18] The PIL contained a detachable reply-slip to complete and return to the researcher (in a reply paid envelope) if they would like to discuss participating in the interview study. Patients being approached to participate in the CGALL trial were asked (for trial purposes) if they would consent to their consultation being audiorecorded. If consent was obtained, these audiorecordings were then analysed for the identification of outcomes.

Focus group participants were identified through the Scottish Health Research register (SHARE - https://www.registerforshare.org/) and sent an invitation letter asking them to contact the research team if interested in participating. Following initial contact, a researcher phoned the interested participants and ensured they were clear about what the study entailed and arranged a suitable time for the focus group.

#### Data collection, management and analysis

One author (RN) conducted the interviews over the telephone between (April– and August, 2017). The focus group was conducted by two members of the trial team (KG and the PPI partner BC) on 20 July 2017. Trial consultations were conducted as standard and four sites across the trial were sampled to inform outcome identification. Informed (written and recorded) consent was obtained from all participants prior to data collection, and confidentiality of the participants was assured. Participants were encouraged to consider what aspects of their disease or treatment impacted them most, both in terms of physical and psychological functioning and what improvements they would wish to see in terms of outcomes. Interviews, focus group and audioconsultations were audiorecorded and transcribed verbatim using a professional transcription service. All transcripts were imported into NVivo (V.10, 2013: QSR International) and analysed using conventional content analysis (ie, coding categories are derived directly from the text data and are used to interpret meaning from the content).[19] Various themes and subthemes were generated by one researcher (RN) based on the contents of the transcripts to identify the outcomes stated by the participants, and these were then further discussed (with KG) to finalise the list of outcomes identified across the primary qualitative data. The analysis was oriented to address the aim of identifying the range of outcomes that might be considered important and the reasons used to justify assessment of them as important.

### Categorisation of identified outcomes into outcome domains

The list of potential outcomes generated from the systematic evidence search and primary qualitative research formed the basis of a 'long' list of outcomes used to refine the items into a final 'short' list for inclusion in the Delphi stage of core outcome set development. Outcomes were first grouped and reduced according to original source, that is, the initial long list from the evidence review was reduced for duplication by two members of the research team (MC and IA). A similar process was conducted to deduplicate the outcomes identified from the PROM coding, qualitative evidence synthesis and primary qualitative research (KG and RN). These outcomes lists were then merged to identify areas of overlap and reduce for further duplication through iterative group discussions (further addressing duplications and relevance of outcomes for effectiveness trials) to produce a final short list OF individual outcomes of relevance in this context.

Individual outcome items were further grouped into broader concept level headings to categorise outcome domains. These concept-level headings were informed by other outcome categorisation work in the area of COS and supplemented through study management group discussion. The categorisation was performed by one member of the team (KG) and refined through iterative discussions.[15 20 21]

### Patient and public involvement

The outcome mapping work reported in this manuscript has involved the input of patients from inception, through design, conduct and reporting. Coauthor BC is a patient partner and has been involved in all phases of project design and delivery and specifically helped to facilitate the patient focus groups to identify outcomes of importance to patients.

## RESULTS

### Studies identified from the search of existing literature

The literature search identified 794 (633 from the search of trials, 60 from the PROM search and 101 from the search of qualitative literature) potentially relevant titles and abstracts for screening. Of these, 137 (129 from trials and PROMs and 8 qualitative studies) were deemed eligible for inclusion (figure 1). A total of 137 publications

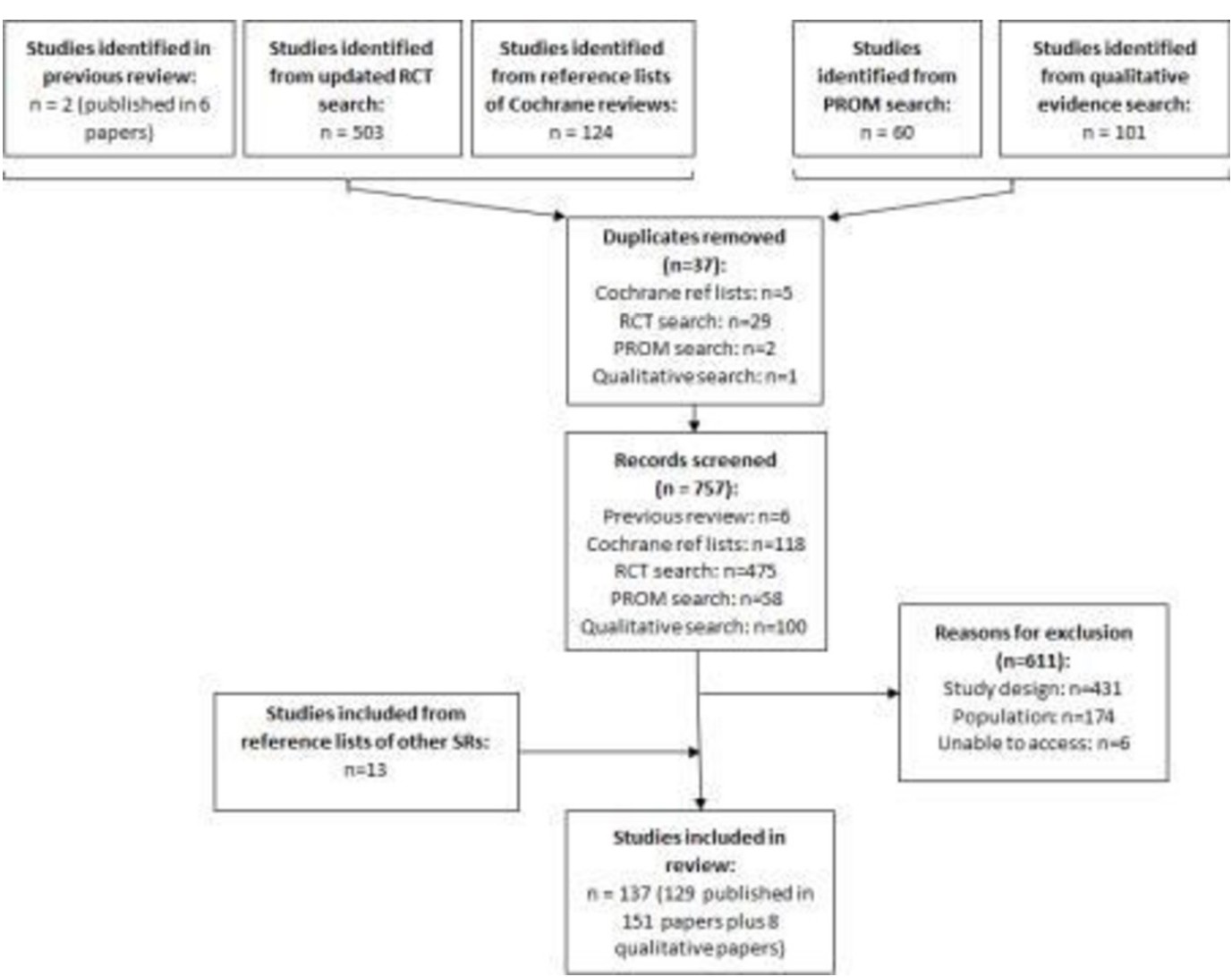

**Figure 1** PRISMA systematic review diagram for evidence synthesis. PRISMA, Preferred Reporting Items for Systematic Reviews and Meta-Analyses; PROMs, patient-reported outcome measures; RCTs, randomised controlled trial; SR, Systematic Reviews.

from 129 RCTs, 4 gallstone disease-specific PROMs and 8t qualitative studies were therefore included

### Sample characteristics of existing literature and new primary research

#### Existing literature

Seven studies were conducted in the UK. A further 44 were conducted in countries in Europe, other than the UK, 12 in the USA and 66 in a range of other countries. A total of over 10 000 participants were included in the studies, predominantly women (sex ratio around 2.5:1) in their mid-40s (40–53, median 46.1). Studies were mainly single-centre (n=113) and ranged in size from small (14 participants) to large (618 participants) trials (median n=75). A vast majority of trials involved one type of surgery vs another type of surgery, with single incision laparoscopic cholecystectomy vs conventional laparoscopic cholecystectomy being the most common trial configuration.[22–71] Other common configurations were early laparoscopic cholecystectomy versus delayed laparoscopic cholecystectomy,[72–87] mini laparoscopic cholecystectomy versus laparoscopic cholecystectomy,[88–97] mini laparotomy versus laparoscopic cholecystectomy[98–103] and day-case laparoscopic cholecystectomy versus overnight stay laparoscopic cholecystectomy.[104–107] There were few non-surgical interventions; one study compared shock wave lithotripsy and laparoscopic cholecystectomy,[108] while several compared observation with cholecystectomy.[109 110] The four disease specific PROMS for uncomplicated gallstone disease were developed in Canada (n=2), .New Zealand (n=1) and Germany (n=1).[111–114] Two of the PROMS focused on gallstone disease,[111 113] one on gastrointestinal diseases,[114] and the other on quality of life after abdominal surgery.[112] All of the PROMs reported to measure multidimensional constructs, for example, quality of life. The PROMs varied in the number of constructs they aimed to assess (ranging from 4-8) and the number of items they asked participants to report on (ranging from 5 to 41, median=27).

**Table 1** Summary characteristics and demographics of included studies

| Characteristic | Quantitative review | Qualitative review |
| --- | --- | --- |
| No of study participants | (n=119 studies) | n=8 studies |
| Median | 75 | 19.5 |
| Range | 14–618 | 6–100 |
| Total | 10 757 | 256 |
| No of males/females | (n=96 studies) | (n=7 studies) |
| Median | 21/53.5 | 4/16 |
| Range | 0–255/15–415 | 2–15/4–37 |
| Total | 2632/6166 | 43/113 |
| Mean age (years) | (n=14 studies) | – |
| Median (years) | 46.1 | – |
| Range (years) | 40–53 | 19–81 |
| Country | (n=128 studies) | (n=8 studies) |
| UK | 7* | 1 |
| European Economic Area (excl UK) | 44* | 3 |
| USA | 12* | 1 |
| Other | 66 | 3 |
| No of centres | (n=129 studies) | (n=8 studies) |
| Single centre | 113 | 8 |
| Multicentre | 16 | |
| Total no of outcomes reported | 330 | 17 |

*One study (Marks 2013) was conducted in the UK, USA and Italy and is included in the count for each country; number of studies (n=xx studies) in table relates to how many studies reported each relevant characteristic.

The eight qualitative studies from seven different countries: USA (n=2),[115 116] UK,[117] Brazil,[118] Canada,[119] Sweden,[120] the Netherland[121] and Spain[122] were identified including 324 participants (ranged from 12 to 162, median=19.5). They were predominantly women and their age ranged from 19 to 81 years. Seven studies used interviews either face to face (n=5) or by telephone (n=2) and one used focus groups for data collection. All of the treatments being explored in the studies were surgery but the types of surgery varied. Five studies investigated patients' experiences after surgery, two investigated experiences of cholecystitis (ie, inflammation of the gall bladder) and surgery and another investigated experiences of cholelithiasis (ie, presence of gallstones).[115–122] Table 1 provides a summary of the characteristics of participants included in the quantitative and qualitative literature review.

## Primary research

Six individual interviews, one focus group (n=5 participants) and analysis of 20 consultations (5 each from 4 different hospital sites) were conducted providing data from a total sample of 31 patients. A brief description of the participants is provided in online supplemental appendix 2. They included 26 women and five men, 26 of whom had been approached to take part in the CGALL trial (12 trial consenters and 14 trial non-consenters), and 5 patients whom had not been approached about the trial but who had all had a cholecystectomy. The CGALL trial is an RCT comparing laparoscopic cholecystectomy with conservative management for preventing recurrent symptoms and complications in adults with uncomplicated symptomatic gallstones.[18]

### Outcomes identified from existing literature and new primary research

A total of 386 individual recorded outcomes were identified across the combined evidence from existing literature and new primary research: 330 outcomes (which were reported 1147 times) from trials evaluating interventions, 22 outcomes from PROMs, 17 outcomes from existing qualitative studies, and 17 outcomes from primary qualitative research.

Of the 330 individual outcomes reported in trials, 97 (29.4%) were reported as 'primary outcomes' with only 64 (19.4%) being formally defined and 227 (68.8%) were reported by one study only. The three 'verbatim' outcomes which were reported most frequently were: 'operating time' (n=103), 'postoperative pain' (n=72) and 'conversion to open surgery' (n=71). In addition, two studies reported the outcome 'conversion'. Four studies reported further outcomes relating to 'operating time', that is, 'anaesthetic time' (n=1), 'surgical time' (n=2), 'procedure time' (n=1). Forty studies reported further outcomes relating to pain: 'umbilical pain' (n=1), 'abdominal pain' (n=9), right upper quadrant pain attacks' (n=3), 'chronic postsurgery pain' (n=1), 'overall pain' (n=2), 'incisional pain' (n=10), 'pain in the context of sexual intercourse' (n=1), 'Visual Analogue Pain Score' (n=2), 'pain' (n=2), 'shoulder pain' (n=4), 'pain or other symptoms in incision area/port sites' (n=1), 'patient reported location of pain at initial postop visit' (n=1), 'site of most severe pain' (n=1), 'admission due to pain' (n=2). Table 2 provides a list, and frequency, of all outcomes reported in the included trials.

Some 106 individual items were identified from the four gallstone disease specific PROMS covering 22 individual outcomes, with frequency of each outcome varying across the individual PROMs (table 3). Included PROMs covered between eight and 14 domains. None of the included PROMS reported whether patients were involved in the measures development. Pain and emotional outcomes were the most frequently covered with 17 items each (making up 32% of total items) and reported across all four PROMs. There were seven outcomes identified only once (and four of which were identified in one PROM) across the 106 items, which included thirst/dehydration, cognitive, service use, body image, sexual function, regurgitation and swallowing.

**Table 2** Outcomes reported in randomised controlled trials of interventions to treat uncomplicated gallstone disease

| Domain | Outcome | No of times reported |
|---|---|---|
| Physical | Physical activity postop | 1 |
| | Recovery of self-reported physical activity | 1 |
| | Time to resume normal physical activity | 3 |
| | Time to resume walking | 3 |
| | Functional impairment | 1 |
| Role | Time to everyday life | 1 |
| | Return to daily activities | 1 |
| | Return to normal function | 1 |
| | Return to activities of daily living | 1 |
| | Days from surgery to normal activity | 3 |
| | Time to return to normal activities | 1 |
| | Days to full activity | 1 |
| | No of days sick leave | 3 |
| | Time away from work | 2 |
| | Loss of active days of work | 1 |
| | Time to return to work after discharge | 1 |
| | Time to return to work | 8 |
| | Days to return to school | 1 |
| | Convalescence time | 1 |
| | Postop regain of functionality | 1 |
| | Total recuperation period | 1 |
| | Mobilisation | 1 |
| Pain | Overall pain | 2 |
| | Pain | 2 |
| | Visual analogue pain score | 2 |
| | Site of most severe pain | 1 |
| | Umbilical pain | 1 |
| | Abdominal pain | 9 |
| | Right upper quadrant pain attacks | 3 |
| | Shoulder pain | 4 |
| | Admission due to pain | 2 |
| | Incisional pain | 10 |
| | Whether had pain or other symptoms in incision area/port sites | 1 |
| | Patient-reported location of pain at initial postop visit | 1 |
| | Postoperative pain | 72 |
| | Chronic post surgery pain | 1 |
| | Analgesic requirements | 58 |
| | Duration of analgesia use | 2 |
| | Time to first analgesics | 1 |
| | Days of medication required | 1 |
| | Analgesia requirement | 1 |
| | No of patients needing additional analgesia | 1 |
| | Use of peripheral analgesics | 1 |
| | Use of centrally acting analgesia | 1 |

Continued

| Domain | Outcome | No of times reported |
|---|---|---|
| | Total pethidine (mg) | 1 |
| | Pethidine requirement | 1 |
| | Suppository requests | 1 |
| Bowel movements | Return of bowel function | 1 |
| | Change in bowel habit | 1 |
| | Diarrhoea or loose stools | 2 |
| | Time to resume passing stools | 1 |
| Thirst/dehydration | Resumption of oral diet within 24 hours | 1 |
| | Resumption of orals | 1 |
| | Resumption of oral intake | 3 |
| | Normal drinking | 1 |
| | Return to liquid feeds | 1 |
| | Time to clear liquids | 1 |
| Appetite/eating/taste | Time to resume eating | 4 |
| | Normal eating | 1 |
| | Return to solid diet | 2 |
| | Time to resume feeding | 1 |
| Fatigue | Fatigue | 2 |
| Sleep | Length of night sleep | 1 |
| | Awakening during the night | 1 |
| Generic health Dietary habits | Overall health state | 1 |
| | Health status | 1 |
| | Satisfaction with life in general | 1 |
| | Patient satisfaction | 2 |
| | Overall satisfaction | 2 |
| | Satisfaction score | 1 |
| | Global patient satisfaction | 1 |
| | Patient overall satisfaction | 1 |
| | Time interval between onset of symptoms and admission to hospital | 1 |
| | Need to avoid fried/fatty foods after surgery | 3 |
| Social | Time away from recreational activity | 1 |
| | Return to going out (days) | 1 |
| Belching/Bloating/Gas | Time to resume passing intestinal gases | 1 |
| | Flatulence and/or dyspepsia | 1 |
| Vomiting/nausea | Nausea or vomiting | 5 |
| | Vomiting | 7 |
| | Nausea | 8 |
| Reflux | Reflux symptoms | 2 |
| Regurgitation | Heartburn or regurgitation | 1 |
| | Heartburn or regurgitation | 1 |
| Body Image | Satisfaction with body image | 3 |
| | Patient derived body image | 1 |
| | Satisfaction with aesthetic result | 1 |
| | Score on body image scale | 2 |
| | Wound satisfaction | 2 |

Continued

**Table 2** Continued

| Domain | Outcome | No of times reported |
|---|---|---|
| | Satisfaction with cosmetic outcome | 10 |
| | Incision satisfaction | 1 |
| | Patient satisfaction score on surgery and scars | 2 |
| | Cosmetic satisfaction of surgical scar | 2 |
| | Patient satisfaction score on scars | 1 |
| | Cosmetic outcome | 13 |
| | Score on cosmetic scale | 1 |
| | Cosmesis | 5 |
| | Cosmetic score | 1 |
| | Cosmetic result | 6 |
| | Incisional cosmesis | 1 |
| | Incision cosmetic result | 1 |
| | Scar evaluation | 4 |
| | Appearance of each incision | 1 |
| Sexual function | Satisfaction in the context of sexual intercourse | 1 |
| | Pain in the context of sexual intercourse | 1 |
| | Dyspareunia | 1 |
| Generic symptoms | Quality of life | 22 |
| | Whether operation had had any impact on Quality of Life | 1 |
| | Morbidity | 8 |
| | Overall morbidity following diagnosis | 1 |
| | General discomfort | 1 |
| | Symptoms | 1 |
| | Symptoms during waiting period | 2 |
| | Residual abdominal symptoms | 3 |
| | Examined for residual symptoms | 1 |
| | Gastrointestinal complaints | 1 |
| | Failure of conservative treatment | 4 |
| | Failure of technique | 1 |
| | No of patients recovered | 1 |
| | Complication rates during waiting time for elective laparoscopic cholecystectomy | 1 |
| | Success rate | 2 |
| Mortality | Mortality | 30 |
| Intraoperative adverse | Common bile duct (CBD) stones | 1 |
| Events | CBD injury | 3 |
| | Biliary leak | 1 |
| | Bile leakage | 2 |
| | Biliary injury | 1 |
| | Bile duct injury | 2 |
| | Bile spillage | 1 |
| | Bile duct lesions | 1 |
| | Haemorrhage | 1 |
| | Blood loss | 30 |
| | Rate of intraoperative bleeding | 1 |
| | Intraoperative bleeding | 1 |

**Table 2** Continued

| Domain | Outcome | No of times reported |
|---|---|---|
| Intraoperative and postoperative | Adverse events | 6 |
| Adverse events | Intraoperative and postoperative complications | 1 |
| | Operative complications | 43 |
| | Complications | 6 |
| | Other complications | 1 |
| | General complications | 2 |
| | Global complications | 1 |
| | Major complications | 1 |
| | Abdominal wall complications | 1 |
| | Port-site complications | 1 |
| | Tissue damage | 1 |
| | Intra-abdominal collections | 1 |
| Postoperative adverse | Postoperative complications | 66 |
| Events | Parietal complications after surgery | 2 |
| | Postoperative infection rate | 1 |
| | Wound infection(s) | 7 |
| | Rate of wound infection | 1 |
| | Surgical site infection | 1 |
| | Incision infection | 1 |
| | Postoperative wound-related infection | 1 |
| | Wound condition | 1 |
| | Postoperative wound-related hernia | 1 |
| | Porthernia | 2 |
| | Postoperative hernia | 1 |
| | Incisional hernia occurrence | 6 |
| | Trocar herniation | 2 |
| | Intra-abdominal adhesions | 1 |
| | Satisfaction with surgery in general | 1 |
| | Patient satisfaction score on surgery | 2 |
| | Satisfaction with operation | 5 |
| | Satisfaction with overall procedural result | 1 |
| | Patient satisfaction after surgery | 1 |
| | Perceived success of operation | 1 |
| | Recommends the procedure | 3 |
| | Patient preference survey including 'willingness to pay for single incision laprascopic cholecystectomy' questionnaire | 1 |
| | Procedure believed to have undergone | 1 |
| Service use | Readmission before elective operation | 1 |
| | Operating time | 103 |
| | Operative data | 2 |
| | Surgical time | 2 |
| | Extensions | 1 |
| | Procedure time | 1 |
| | System setup time | 1 |
| | Performance time | 1 |

Continued

| Domain | Outcome | No of times reported |
|---|---|---|
| | Duration of each operative stage | 1 |
| | Learning curve for operating time | 1 |
| | Duration of admission | 1 |
| | Hospital stay | 60 |
| | Postoperative hospital stay | 23 |
| | Day surgery | 2 |
| | Successful completion of day surgery | 3 |
| | Reasons for overnight stay in patients scheduled for day surgery | 1 |
| | Post Anaesthesia Care Unit length of stay | 1 |
| | Discharge >24 hour postop | 1 |
| | Discharge from hospital | 1 |
| | Discharge from hospital 20–24 hours postoperative | 1 |
| | Time until discharge | 1 |
| | Time from operation to discharge | 1 |
| | Grade of surgeon | 1 |
| | Grade of operating surgeon | 1 |
| | Resident's participation | 1 |
| | Need to contact hospital or other healthcare providers after discharge | 2 |
| | Ambulatory rate | 1 |
| | No of patients requiring readmission to hospital | 5 |
| | Readmission | 3 |
| | Causes of hospitalisation | 2 |
| | No of patients requiring reintervention | 2 |
| | Additional procedures | 1 |
| | Reoperation | 1 |
| | Reintervention required | 1 |
| | Revision surgery | 1 |
| | Conversion to open surgery | 71 |
| | Conversion of 5–10 mm port | 2 |
| | Modification of operative technique | 4 |
| | Conversion to Laprascopic cholesytectomy (LC) | 16 |
| | Conversion to Laproendoscopic single site procedure | 1 |
| | Conversion to other laparoscopic approach | 4 |
| | Conversion | 2 |
| | Conversion from Single Incision Laprascopic Cholecystectomy (SILC) to 4PLC (Four Port Laprascopic Cholecystectomy) | 2 |
| | Rate of cholecystectomy | 2 |

**Table 2** Continued

Continued

| Table 2 | Continued | |
|---|---|---|
| **Domain** | **Outcome** | **No of times reported** |
| Cost-effectiveness | Cost of operation | 6 |
| | Cost analysis | 1 |
| | Procedural cost | 1 |
| | Total encounter cost | 1 |
| | Hospital cost | 5 |
| | Hospital charges | 1 |
| | Total cost | 1 |
| | Costs | 1 |
| | Charge data | 1 |
| | Cost-effectiveness ratio | 1 |
| | Economic analysis of early versus conventional management for newly diagnosed GB disease | 1 |
| Process (operation or trial) | Pulmonary function | 5 |
| | Spirometric indices | 1 |
| | Total carbon dioxide insufflation (litres) | 1 |
| | Surgical stress response | 2 |
| | No of cannulas used | 1 |
| | Pressure of the pneumo peritoneum | 1 |
| | Heart rate variability | 1 |
| | Serum cytokines | 1 |
| | Serum interleukin-6 | 3 |
| | Serum C reactive protein (CRP) | 3 |
| | Blood count | 1 |
| | Liver function tests | 1 |
| | Alpha-defensins expression | 1 |
| | hsCRP values | 1 |
| | CRP values | 1 |
| | Electroconductivity of representative dermatones | 1 |
| | Central & peripheral temperature | 1 |
| | Blood examination | 1 |
| | Histological findings | 1 |
| | Histopathological findings | 1 |
| | Histopathological diagnosis | 1 |
| | Occlusive bandages in place | 1 |
| | General anaesthesia | 1 |
| | Anaesthetic time | 1 |
| | Amount of bupivacaine used | 1 |
| | Length of skin incision | 9 |
| | Extra skin incisions required | 1 |
| | Aponeurosis wound size | 1 |
| | Wound length | 3 |
| | Scar length | 2 |
| | Intraoperative cholangiography performed | 3 |
| | Intraoperative technical performance | 1 |
| | Intraoperative diagnosis | 1 |
| | Intraoperative findings | 3 |

**Table 2** Continued

| Domain | Outcome | No of times reported |
|---|---|---|
| | Postop metabolic and hormonal levels | 1 |
| | Postoperative level of 8-epiPGF2$\alpha$ | 1 |
| | Postoperative level of uric acid | 1 |
| | Postoperative forced expiratory volume in 1 s | 1 |
| | Change of antibiotic therapy due to nonresponse or To intolerance of moxifloxacin | 1 |
| | Use of rescue therapy with 125 mg lysine clonixinate Tablets | 1 |
| | Time of surgical dissection | 1 |
| | Time of closure | 1 |
| | Detailed surgical time course | 1 |
| | Rate of operative error | 1 |
| | Technical problems | 2 |
| | No of admissions between 8:00 and 17:00 hour | 1 |
| | Admission on Monday to Thursday | 1 |
| | Difficulty of case | 2 |
| | Operative difficulty | 1 |
| | Surgical difficulty | 1 |
| | Difficulty of dissection | 1 |
| | Difficulty (impaired) of exposure | 1 |
| | Prediction of Laprascopic Cholecystectomy (LC) difficulty | 1 |
| | Surgeon's perceptions of difficulties during operation | 1 |
| | Potential for increased surgical risk to the patient | 1 |
| | Critical view of safety | 1 |
| | Assessment of surgical handling | 1 |
| | Feasibility and safety of SILC versus 4PLC | 1 |
| | Ultrasonography findings | 1 |
| | No of stones | 1 |
| | Size of largest stone | 1 |
| | Rate of GB rupture | 1 |
| | GB wall thickness | 1 |
| | Time to retrieve GB | 1 |
| | No and type of instruments | 1 |
| | Use of extra umbilical rescue device 2.3 mm mini-LC Instrument | 1 |
| | Insertion points | 1 |
| | No of incisions | 1 |
| | Port sizes | 1 |
| | Additional ports required | 9 |
| | Port enlargement for GB removal | 1 |
| | Additional trocars used | 2 |
| | Fourth trocar added | 2 |
| | Trocar use | 1 |
| | Placement of any additional laparoscopic ports other than the Single Incision and Transanal Surgery (SILS) port | 1 |

Continued

**Table 2** Continued

| Domain | Outcome | No of times reported |
|---|---|---|
| | Time required for insertion of SILS port system compared with four standard ports | 2 |
| | Success rate (3-port vs 4-port LC) | 2 |
| | Need to amplify extraction port | 2 |
| | No of ports used | 1 |
| | Reason for port placement | 1 |
| | Time of trocar introduction | 1 |
| | Use of Keith needle | 1 |
| | Successful completion of needlescopic operation | 1 |
| | No of drains placed | 1 |
| | Requirement of drains | 1 |
| | No of manual camera corrections | 1 |
| | No of cleaning of the optics | 1 |
| | Ability to achieve optimal focus on the operative site During the procedure | 1 |
| | Positioning accuracy of image | 1 |
| | No of clearing camera | 1 |
| | Time of adjusting operative field | 1 |
| | Surgeon's comfort | 1 |
| | Comfort of instrument positioners | 1 |
| | No of actions in positioning of laparoscope | 1 |
| | Feasibility of surgical procedures using only the camera robot | 1 |
| | Commands misunderstood | 1 |
| | No of commands issued by the surgeon | 1 |
| | Sum of operative actions | 1 |
| | No of dissection actions | 1 |
| | No of grasping actions | 1 |
| | Time from admission to ultrasonography | 1 |
| | Time from ultrasonography to recruitment | 1 |
| | Time from recruitment to operation | 1 |
| | Time from recruitment to discharge | 1 |
| | Completion of randomised treatment | 2 |
| | Gallstone-associated events after randomisation | 2 |

Seventeen individual outcomes were identified from the existing qualitative literature (see table 4). Twelve overlapped with existing outcomes identified in the literature and the PROMS leaving five additional outcomes for consideration in the long list, namely: dizziness, fainting, trust, weight and prevention of additional disease.

The primary qualitative research identified 17 individual outcomes, with the majority (n=14) overlapping with those reported in the previously reviewed evidence (See table 4). However, three additional outcomes (breathing problems, cough and mortality) were identified that were not in the previous patient focused evidence (PROMs and qualitative literature), with two of these (breathing problems, cough) making unique contributions to the overall outcome list.

The 390 individual reported outcomes across the 4 data sources were reduced into a 'short' list of outcomes which could be measured in comparative effectiveness trials (ie, phase III pragmatic evections trials) of interventions to treat uncomplicated symptomatic gallstone disease (see table 5). This resulted in several outcomes being dropped from the long list as deemed not eligible as clinical endpoint outcomes for use in trials of this type (eg, system and process outcomes such as duration of surgery which might be important in earlier phase trials). Therefore, the final list covered 27 broad outcome domains

**Table 3** Inclusion of domains across included PROMs ranked by frequency of domains

| Domain (n=22) | No items per domain | ASIS[112] | CSQ[113] | GIC[111] | GIQLI[114] | Total |
|---|---|---|---|---|---|---|
| Emotional | 17 | ✓ | ✓ | ✓ | ✓ | 4 |
| Pain | 17 | ✓ | ✓ | ✓ | ✓ | 4 |
| Role | 11 | ✓ | ✓ | | | 2 |
| Bowel movements | 8 | ✓ | | | ✓ | 2 |
| Belching/bloating/gas | 7 | | | ✓ | ✓ | 2 |
| Appetite/eating/taste | 6 | ✓ | | ✓ | ✓ | 3 |
| Dietary habits | 6 | | ✓ | ✓ | | 2 |
| Physical | 6 | ✓ | | | ✓ | 2 |
| Fatigue | 5 | ✓ | ✓ | ✓ | ✓ | 4 |
| Social | 4 | | ✓ | ✓ | ✓ | 3 |
| Sleep | 3 | ✓ | | | ✓ | 2 |
| Stomach problems | 3 | | ✓ | ✓ | | 2 |
| Generic health | 2 | | ✓ | | ✓ | 2 |
| Reflux | 2 | | | ✓ | | 1 |
| Vomiting/nausea | 2 | | | ✓ | | 1 |
| Body image | 1 | | | | ✓ | 1 |
| Cognitive | 1 | ✓ | | | | 1 |
| Dysphagia/swallowing | 1 | | | | ✓ | 1 |
| Regurgitation | 1 | | | | ✓ | 1 |
| Service use | 1 | | | ✓ | | 1 |
| Sexual function | 1 | | | | ✓ | 1 |
| Thirst/dehydration | 1 | ✓ | | | | 1 |
| Total | 106 | 10 | 8 | 11 | 14 | |

ASIS, Abdominal Surgery Impact Scale; CSQ, Otago gallstone condition specific questionnaire; GIC, Gallstone Impact Checklist; GIQLI, Gastrointestinal Quality of Life Index; PROMs, patient-reported outcome measures.

that contained 41 distinct outcomes. The domains of pain, intra-operative complications, and post-operative complications contained the most outcomes (n=4 each).

## DISCUSSION

Currently, there is a lack of consistency in the selection, measurement and reporting of outcomes for uncomplicated gallstone disease. This leads to challenges in evidence synthesis and decision-making. A Core Outcome Set would be an important step to improve this situation. This paper which describes an outcome mapping exercise is the first comprehensive step in the development of a core outcome set. It catalogues and reports the outcomes that have been measured in trials of interventions to treatment uncomplicated gallstone disease. It extends this initial phase of outcome identification to include outcomes from PROMs, published qualitative evidence and empirical qualitative research. Over 1000 verbatim outcomes were identified and reduced through deduplication to 390. This was further reduced to 41 outcomes spanning 27 domains. The next steps in this work are to

reach consensus for the COS for uncomplicated symptomatic gall stone disease.

As with many other outcome mapping exercises, this first stage of this study highlights the significant heterogeneity that exists within clinical trials comparing treatments for gallstone disease. Of the 334 outcomes, which were reported multiple times across the 129 RCTs, almost 70% were reported by only one study—a finding comparable with other outcome mapping studies.[21 123 124] All of the effort into collection and reporting of these outcomes is likely wasted as it is doubtful that they could be combined with others to make more confident assessment of the effectiveness of treatments. This outcome heterogeneity in existing trials is further emphasised when considering the outcome of pain, which is reported in 72 trials as postoperative pain but also reported in a number of other trials using 15 different outcomes. The four disease specific PROMs identified further extend the problem of outcome heterogeneity. While all of these measures report to capture quality of life, there is variability in both the inclusion and emphasis of domains

**Table 4** Outcomes reported in PROMs, qualitative evidence synthesis and primary qualitative research

| Outcome | PROMs | Evidence synthesis | Primary research |
|---|---|---|---|
| Emotional | ✓ | ✓ | ✓ |
| Pain | ✓ | ✓ | ✓ |
| Role | ✓ | ✓ | |
| Bowel movements | ✓ | ✓ | ✓ |
| Belching/bloating/gas | ✓ | ✓ | ✓ |
| Appetite/eating/tTaste | ✓ | | |
| Dietary habits | ✓ | ✓ | ✓ |
| Physical | ✓ | | |
| Fatigue | ✓ | ✓ | ✓ |
| Social | ✓ | ✓ | ✓ |
| Sleep | ✓ | ✓ | ✓ |
| Stomach problems | ✓ | ✓ | ✓ |
| Generic health | ✓ | | |
| Reflux | ✓ | ✓ | ✓ |
| Vomiting/nausea | ✓ | ✓ | ✓ |
| Body Image | ✓ | | |
| Cognitive | ✓ | | |
| Dysphagia/swallowing | ✓ | | |
| Regurgitation | ✓ | | |
| Service use | ✓ | | ✓ |
| Sexual function | ✓ | | |
| Thirst/dehydration | ✓ | | |
| Dizziness | | ✓ | |
| Fainting | | ✓ | |
| Prevention of additional disease | | ✓ | |
| Trust | | ✓ | |
| Weight | | ✓ | |
| Cough | | | ✓ |
| Mortality | | | ✓ |
| Problems with breathing | | | ✓ |
| Total | 22 | 17 | 17 |
| Total individual outcomes | | 30 | |

PROMs, patient-reported outcome measures.

captured across each of the tools. For example, the most frequently covered domains capture emotion and pain and are reported across all four tools. Fatigue is also reported across all four tools but is only captured by five

items as opposed to the 17 each for emotion and pain. Nine of the 22 domains are only captured by one of the PROMs, with a further eight domains being captured by 2 PROMs. Again this variability in what individual items are measured in these tools and how they contribute to overall 'quality of life' assessments, raises questions about the legitimacy of combining these measures when evaluating intervention effectiveness. In addition, the relevance of these outcomes to patients must be called into question given the lack of reporting of input from patients in the item inception phase of PROM design across these measures. Two reviews published after completion of this work conducted an methodological assessments of both disease specific and generic PROMs for laparoscopic cholecystectomy and both report considerable variation and a lack of patient involvement.[125 126] There are now several reviews of PROMs in other clinical specialties that also provide findings which further highlight the heterogeneity that exists across measures which, on the surface, report to measure similar concepts.[20 127–129]

The different evidence sources contributed to the final outcome short list in a variety of ways with the outcomes reported in previous trials often capturing clinically focused endpoints and the PROMS and qualitative research providing more patient focused outcomes. When considering what outcomes matter to patients and how this contributes to the outcome mapping in this area, this study used two approaches to ensure patient relevant outcomes were included. The qualitative evidence synthesis and primary research identified a total of 34 combined (of which 21 were mutually exclusive) outcomes, that were important to patients in terms of their gallstone disease or their perceptions about treatments. Theses outcomes could be broadly grouped into physical and social functioning with most reports from participants focusing on reduction in pain and a desire to 'return to normal'. When compared with the outcome domains reported in the PROMs, there was considerable overlap between the two sources. However, there were some areas of discordance between the different sources, with the qualitative data adding a further eight outcomes. In addition the qualitative evidence synthesis and the empirical research identified outcomes not previously measured or reported in comparative effectiveness trials for uncomplicated gallstone disease. This value of including evidence from existing literature exploring patients perspective and/or new primary research to identify patient relevant outcomes is gaining traction among COS developed.[5] Where most have included outcomes through identification in interviews, the use of qualitative evidence synthesis is growing, and especially in areas where there has previously been a considerable volume of work to draw on.[130] These studies have shown that this work contributes previously unreported outcomes that are of importance to patients and hence underpins the critical nature of this step in COS development. Whether these outcomes identified in these list development stages end up making it into the COS

**Table 5** Outcomes map for uncomplicated gallstone disease

| Outcome domains | Outcome types | Definitions | Contributing evidence source |
|---|---|---|---|
| Physical | Physical activity | Activities such as walking, running, swimming, cycling, physical labour, climbing stairs, gardening, etc | 1, 2 |
| | Exercise | Being able to do activities requiring physical effort, carried out to sustain or improve health and fitness (strength and endurance) | 3 |
| Role | No of days sick leave | Length of time off work after the operation in days | 1,2,3 |
| | Time to everyday life | Length of time taken to return to usual everyday activities | 1,2,3,4 |
| | Impact on others | Impact of your gallstone condition or your gallstone surgery on relationships with people surrounding you | 2,3,4 |
| Pain | Overall pain | Overall pain | 1,2,3,4 |
| | Abdominal pain | General pain occurring at rest and/or when coughing, originating in the abdominal area | 1,2,3,4 |
| | Umbilical pain | Pain around the belly button scar (this is where the main port is that removed the stones) | 1 |
| Bowel movements | Shoulder pain | Pain relating to or affecting the right shoulder region | 1, 2 |
| | Diarrhoea | Watery stools, loose bowel motion | 1,2,3,4 |
| | Constipation | Difficulty passing stool | 2,3,4 |
| Thirst/dehydration | Resumption of orals | Starting to eat and drink after treatment | 1, 2 |
| Appetite/eating/taste | Time to resume eating | Length of time taken to return to oral food intake | 1,2,3,4 |
| Fatigue | Fatigue | Feeling physically or mentally tired or lacking in energy | 1,2,3,4 |
| Sleep | Length of night sleep | Length of night's sleep | 1,2,3,4 |
| Cognitive | Difficulty concentrating | Inability to focus attention on one task or problem | 2 |
| Emotional | Anxiety | A feeling of worry, nervousness or unease | 2,3,4 |
| | Distress | A feeling of extreme anxiety, stress or anguish | 2,3,4 |
| | Trust | belief in the reliability, truth or ability of someone or something | 3 |
| Generic health | Quality of life | How well you feel physically and emotionally because of a combination of: your gallstones | 1,2,3,4 |
| | | The prospect of treatment | |
| | | The result of treatment (Treatment might include surgery or painkillers) | |
| | Overall health state | Overall state of your physical and mental condition | 1 |
| | Overall satisfaction | The degree to which expectations or needs have been fulfilled | 1 |
| Dietary habits | Food intolerance | A physical adverse reaction by the body to certain foods | 1, 4 |
| Social | Time away from recreational activities | Time spent away from enjoyable activities as a result of your gallstone condition or gallstone surgery | 1,2,3,4 |

**Table 5** Continued

| Outcome domains | Outcome types | Definitions | Contributing evidence source |
|---|---|---|---|
| Belching/bloating/gas | Flatulence | Belching, farting, bloating or gas | 1,2,3,4 |
| | Bloating | Abdominal swelling as a result of excess fluid or gas | 2,3,4 |
| | Abdominal discomfort | Pain or discomfort in the stomach area | 2,3,4 |
| Vomiting/nausea | Vomiting | Being sick | 1,2,3,4 |
| | Nausea | Feeling sick | 1,2,3,4 |
| Reflux | Heartburn | A form of indigestion that presents as a burning sensation in the chest, caused by acid reflux | 1,2,3,4 |
| Body image | Satisfaction with body image | A feeling of satisfaction with your own physical appearance | 1, 2 |
| | Satisfaction with outcome | The extent to which you are content with the cosmetic results of gallstone surgery | 1, 2 |
| Sexual function | Satisfaction in the context of sexual intercourse | The extent to which you are satisfied with experiences of sexual intercourse in relation to your gallstone condition or your gallstone surgery | 1, 2 |
| | Pain in the context of sexual intercourse | The extent to which you are experiencing pain during or after sexual intercourse in relation to your gallstone condition or your gallstone surgery | 1, 2 |
| Regurgitation | Regurgitation | Bringing swallowed food back up to the mouth | 1, 2 |
| Dysphagia/swallowing | Trouble swallowing food | Problems swallowing food | 2 |
| Generic symptoms | General discomfort | An unpleasant feeling and/or low-level pain which is hard to define | 1,2,3,4 |
| | Residual symptoms | Continuing to have symptoms (such as pain, bloating, etc) after removal of the gallbladder | 1,2,3,4 |
| | Dizziness | Feeling light headed or dizzy | 3 |
| | Fainting | Fainting (short-term loss of consciousness) | 3 |
| Mortality | Mortality | Death from any cause | 1,4 |
| Intraoperative adverse events | Common bile duct stones | Stones in the common bile duct | 1 |
| | Common bile duct injury | During surgery the common bile duct is damaged | 1 |
| | Biliary leak | Liver produces bile which is stored in the gallbladder (see diagram). If this is damaged the bile can leak and cause complications. | 1 |
| | Haemorrhage | Bleeding or the abnormal flow of blood; the release of blood from a ruptured blood vessel | 1 |
| Intra and post-operative adverse events | Intra-abdominal collections | After surgery any type of fluid collecting in the abdomen. | 1 |

**Table 5**  Continued

| Outcome domains | Outcome types | Definitions | Contributing evidence source |
|---|---|---|---|
| Postoperative adverse events | Hernia occurrence | internal hernia—displacement of an organ within the abdomen through a potential defect. | 1 |
| | Port-site complications | Complications such as infection, hernia, pain or bleeding at or within the 'keyholes' characteristic of keyhole surgery | 1 |
| | Wound infections | An infection at the wound site | 1 |
| | Patient perceived success of operation | How patient perceive the success of the operation | 1, 3 |
| Service use | Hospital stay | Length of time spent in the hospital from admission to discharge | 1, 2,4 |
| Cost-effectiveness | Hospital cost | Total hospital costs, taking into account the total length of hospital stay, operating room charges, medical and surgical supplies, pharmacy, laboratory and pathology, recovery room, anaesthesia and Intensive care unit/ observation rooms | 1 |
| | Overall cost | Cost of use of healthcare services; for example, contact with a GP, in- or outpatient contact, prescribed medications | 1 |
| | Cost-effectiveness ratio | Cost-effectiveness of treatment route (medical management or surgery to remove the Gallbladder), calculated by dividing cost by success rate (defined by the quality of life after treatment) | 1 |

1. Trials of interventions for gallstone disease.
2. Patient reported outcome measures.
3. Qualitative evidence synthesis.
4. Primary qualitative research.
GP, general practitioner.

is currently less well evidenced but will be important to know.

## Strengths and limitations

This outcome mapping exercise used a systematic search to identify outcomes reported in both quantitative and qualitative studies in the literature. In addition to this rigorous systematic search we supplemented the pool of outcomes already available with new primary qualitative research (through three methods) to further identify outcomes that matter to patients who are experiencing uncomplicates symptomatic gallstone disease. This complementary approach to identification of outcomes has ensured a broad catalogue of both clinically relevant and patient important outcomes. Limitations of this review are linked to the inclusion of only English language studies, a lack of quality appraisal of included studies, and no assessment of reporting bias. While these decisions were fit for purpose for the COS activity, they may have introduced potential reporting and selection bias within the outcome map. With regard to outcome reporting bias, other COS development papers have explored this and found that in surgical studies of oesophagectomy and colorectal cancer resection papers frequently did not report all the outcomes intended to be measured (50% at least or more did not do that).[131 132] Future COS development studies should consider this approach to assess outcome reporting bias. It would also have been useful to collect the study teams reported rationale for the selection of reported outcomes to determine how that process was determined.

## CONCLUSIONS

This study took a rigorous approach to catalogue and map the outcomes of importance in gallstone disease to enhance the development of the COS 'long' list. The synthesis of data from the four different evidence sources further underpinned the need for a COS in this space due to the heterogeneity of outcome measurement and reporting. However, the extensive use of data sources to contribute to the development of the list of outcomes for further consensus agreement, did highlight 'new' outcomes that have not been previously reported for trials evaluating interventions for gallstone disease and many of these 'new' outcomes were those reported by patients. This comprehensive approach to the development of the long list, and then ultimately the short list for scoring in a COS, gives confidence that both clinically relevant and patient focused outcomes have been considered and have the potential to be represented in the agreed COS.

**Author affiliations**

[1]Health Services Research Unit, University of Aberdeen Institute of Applied Health Sciences, Aberdeen, UK

[2]Department of Social Medicine, University of Bristol Department of Social Medicine, Bristol, UK

[3]Department of Surgery, NHS Grampian, Aberdeen, UK

[4]Department of Surgery, ElKabbary Hospital, Alexandria, Egypt

[5]Clinical Biochemistry, Grampian University Hospitals NHS Trust, Aberdeen, UK

**Contributors** Conceptualisation: CR. Methodology: MC, RN, JB, IA, MBr, CR, KG. Investigation MC, RN, JB, MB, BC, KG. Formal Analysis MC, RN, JB, IA, MB, CR and KG. Writing-original draft: MC, RN and KG. Writing-review and editing: MC, RN, JB, IA, MBe, KI, CR and KG. Visualisation: MC, RN and KG. Supervision: KG. Project Administration: MC, RN, KI and KG. Funding acquisition: JB, IA, MBr, BC, CR and KG.

**Funding** This work was supported by the National Institute for Health Research (NIHR) Health Technology Assessment (HTA) Programme grant (14/192/71). The work was also supported by an NHS Grampian Endowment grant (16/11/006). KG held a Medical Research Council UK Methodology Fellowship during the delivery of this project (MR/L01193X/1). The Health Services Research Unit, Institute of Applied Health Sciences (University of Aberdeen), is core funded by the chief scientist office of the Scottish Government Health and Social Care Directorates.

**Disclaimer** The funders had no involvement in study design, collection, analysis and interpretation of data, reporting or the decision to publish.

**Competing interests** None declared.

**Patient consent for publication** Not required.

**Ethics approval** The primary qualitative research in this study was approved as part of the CGALL trial by the North of Scotland Research Ethics Service (16/NS/0053) and NHS Grampian Research and Development.

**Provenance and peer review** Not commissioned; externally peer reviewed.

**Data availability statement** All data relevant to the study are included in the article or uploaded as online supplemental information. No further data are available.

**ORCID iDs**
Jane Blazeby http://orcid.org/0000-0002-3354-3330
Miriam Brazzelli http://orcid.org/0000-0002-7576-6751
Bernard Croal http://orcid.org/0000-0002-6375-3507
Karen Innes http://orcid.org/0000-0001-8512-4368
Craig Ramsay http://orcid.org/0000-0003-4043-7349
Katie Gillies http://orcid.org/0000-0001-7890-2854

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
