## [Reviewer comments · BMJ Open]

ARTICLE DETAILS

TITLE (PROVISIONAL)	Identification and categorisation of relevant outcomes for symptomatic uncomplicated gallstone disease: in depth analysis to inform the development of a core outcome set
AUTHORS	Cruickshank, Moira; Newlands, Rumana; Blazeby, Jane; Ahmed, Irfan; Bekheit, Mohamed; Brazzelli, Miriam; Croal, Bernard; Innes, Karen; Ramsay, Craig; Gillies, Katie

VERSION 1 – REVIEW

REVIEWER	de Reuver, Philip Radboud university. Nijmegen, The Netherlands
REVIEW RETURNED	16-Nov-2020

GENERAL COMMENTS	With great interest, I read the article entitled: Identification and characterization of relevant outcomes for symptomatic uncomplicated gallstone disease: in-depth analysis to inform the development of a Core Outcome Set. This research group has taken the painful task of compiling all the existing literature on uncomplicated gallstone disease to construct a Core outcome Set for patients with this disease. It is of great clinical importance that this COS will be constructed as the patient group is large and the available literature is mainly focussed on the operation but not on the disease nor the patient-reported outcomes after surgery. After reading the paper in BMJ in October last year, I started looking out for the COS which the authors promised to be a part of the C-Gall project. Unfortunately, this paper is only the result of the literature search and does not yet describe the most relevant outcomes for this specific patient group. Again, I read it with great interest, but the manuscript might be of less interest to the clinician treating these patients compared to the ones involved in the research. I have studied all outcome measures that are summarized in the present study. It is of interest that all that most papers mainly focused on surgical procedures, blood loss, conversion rates, duration of surgery and far less focussed on the patient, the primary stakeholder. Therefore, tables 3 and 4 contributed most information. Interestingly only one item on sick leave was mentioned and only three times reported. This is of my interest as mainly healthy, young and working people represent this population. I think the long list of table 1 could be summarized or published as supplementary. Considering the authors, I am not concerned about the methodological quality of the manuscript. I also see the importance of publication because it is relevant to inform all stakeholders about the relevant outcomes of the treatment for patients with
--

	uncomplicated gallstone disease. It is up to the editor if BMJ open is the most suitable journal for publication. I only have a few comments: please revise the abstract/background and early state that the paper concerns a COS for gallstone patients. Are reference 111 and 113 the only PROMS focussed on gallstone disease? Daliya et al, PMID: 31245403, gives a nice overview of the available PROMS, and the lack of good ones. It could be argued that the Gallstone symptom questionnaire from Vetrhus et al, should be added. Unfortunately, the literature search is restricted until May 2016, 4,5 years ago. Quite some interesting data on this subject came out more recently and would be of value to include in the future COS. While the methodology section describes the qualitative research broadly (line204-226) and triggers the reader on the interviews, the result section is quite briefly on that subject while it is probably of great importance for a well-constructed COS. Appendix 2 only show some patient characteristics, and the result section does not clearly state what is of most importance to most patients with gallstone disease.
--	--

REVIEWER	Deckert, Stefanie Faculty of Medicine Carl Gustav Carus, TU Dresden, Center for Evidence-based Healthcare
REVIEW RETURNED	26-Nov-2020

GENERAL COMMENTS	Thank you for the opportunity to review this paper. I think it is very well designed, conducted, and written. I think the methods were particularly well done and well described. I have only a couple of comments: Referring to studies used to extract endpoints:  - Did any study describe the rationale for the selection of the reported outcomes? It is possible that some trialists have already deliberately dealt with the selection of the outcomes and this would be a relevant information here. - Did you consider the 'outcome reporting bias' when extracting the outcomes? Possibly, for the discussion and your intended process it could be interesting to know which outcomes were measured but not reported. The list of non-reported outcomes could be discussed in light of methodological and content-related aspects within the whole COS team Referring to existing PROMS:  - Did you analyze the development of these tools (cf. inauguration article)? According to COSMINs content validity tool, an excellent PROMs used patients' feedback to evaluate e.g. the relevance of items which are intended to include/which were included in a PROM. It could be a valuable resource to know whether or not patients were already involved in the development and how the relevance was rated. Referring definition of outcomes:  - On page 6 you describe that "any definition of outcomes" was extracted. I am confused because I don't understand how you harmonized the huge mess of reported outcome definitions. Please describe your approach in more detail. Others:  - Page 9: "Participants were encouraged to consider what aspects of their disease or treatment impacted them most, both in terms of physical and social functioning and what improvements they would wish to see in terms of outcomes". Why were psychological aspects not addressed?
--

	- Page 12: “This resulted in several outcomes being dropped from the long list as deemed not eligible as clinical endpoint outcomes for use in trials of this type (e.g. system and process outcomes such as duration of surgery which might be important in earlier phase trials).” Why did you put so much effort into the extraction of outcomes instead of using strict inclusion criteria from the beginning of your review? Good luck for your Delphi study!
--	--

VERSION 1 – AUTHOR RESPONSE

Reviewer #1:		Response
1	With great interest, I read the article entitled: Identification and characterization of relevant outcomes for symptomatic uncomplicated gallstone disease: in-depth analysis to inform the development of a Core Outcome Set. This research group has taken the painful task of compiling all the existing literature on uncomplicated gallstone disease to construct a Core outcome Set for patients with this disease. It is of great clinical importance that this COS will be constructed as the patient group is large and the available literature is mainly focussed on the operation but not on the disease nor the patient-reported outcomes after surgery.	Thank you for this feedback on the importance of the COS.
2	After reading the paper in BMJ in October last year, I started looking out for the COS which the authors promised to be a part of the C-Gall project. Unfortunately, this paper is only the result of the literature search and does not yet describe the most relevant outcomes for this specific patient group. Again, I read it with great interest, but the manuscript might be of less interest to the clinician treating these patients compared to the ones involved in the research.	The reviewer is correct that this paper reports the literature search and synthesis (of both quantitative and qualitative literature) to identify outcomes already reported in the literature. This was supplemented with a content analysis of existing patient reported outcome measures for gallstone disease and primary qualitative research) interviews and focus groups) with patients who have had a diagnosis of gallstones. The paper does identify outcomes that patients have reported as important but it does not identify which of these should be considered core for all evaluations in this setting. This outcome characterisation and mapping stage is a key first step in the development of a core outcome set. The final core outcome set manuscript is in preparation.
3	I have studied all outcome measures that are summarized in the present study. It is of interest that all that most papers mainly focused on surgical procedures, blood loss, conversion rates, duration of surgery and far less focussed on the patient, the primary stakeholder. Therefore, tables 3 and 4 contributed most information.	We agree with the reviewer and this is largely due to the majority of trials in this area comparing different types of surgery rather than surgery to other non-surgical interventions, or head-to-head comparison of non-surgical interventions. This reviewer also recognises the value that Table 3 and 4 bring to the COS development by

		contributing outcomes to the list for scoring that may not have been measured or reported in these trials to date.
4	Interestingly only one item on sick leave was mentioned and only three times reported. This is of my interest as mainly healthy, young and working people represent this population. I think the long list of table 1 could be summarized or published as supplementary.	We believe it is important for the long list to be made transparent and this is in keeping with good reporting practice or COS development as per the COS-STAR guidance (Kirkham JJ et al. Core Outcome Set-STAndards for Reporting: The COS-STAR Statement. PLoS Med. 2016 Oct 18;13(10):e1002148) and other papers reporting mapping exercises as part of the COS development process (Macefield, R.C., et al Developing core outcomes sets: methods for identifying and including patient-reported outcomes (PROs). Trials 15 , 49 (2014). and Hopkins, J C et al. Outcome reporting in bariatric surgery: an in-depth analysis to inform the development of a core outcome set, the BARIACT Study Obesity reviews. , 2015, Vol.16(1), p.88-106). Therefore, we propose to keep it within the main manuscript.
5	Considering the authors, I am not concerned about the methodological quality of the manuscript. I also see the importance of publication because it is relevant to inform all stakeholders about the relevant outcomes of the treatment for patients with uncomplicated gallstone disease. It is up to the editor if BMJ open is the most suitable journal for publication.	Thank you for this feedback.
6	I only have a few comments: please revise the abstract/background and early state that the paper concerns a COS for gallstone patients.	We have edited the objective section of the abstract and background to specify the COS is for patients with gallstones. See page 2 line 26 and page 3 line 65.
7	Are reference 111 and 113 the only PROMS focussed on gallstone disease? Daliya et al, PMID: 31245403, gives a nice overview of the available PROMS, and the lack of good ones. It could be argued that the Gallstone symptom questionnaire from Vethrus et al, should be added.	Thank you for highlighting this review by Daliya et al 2018. This review was published after our search and therefore was not identified in our search. The review has a different focus to this work as it assessed the methodological quality, rather than item content, of both disease specific and generic quality of life (QoL) PROMs. We also identified a more recent review, Alexander et al 2019 (Alexander HC, Nguyen CH, Moore MR, Bartlett AS, Hannam JA, Poole GH, Merry AF. Measurement of patient-reported outcomes after laparoscopic cholecystectomy: a systematic review. Surg Endosc. 2019 Jul;33(7):2061-2071.) which again explored both disease specific and generic QoL measures. We have now highlighted these reviews in the discussion section of the paper. See page 12 line 409-412.

		We would want to avoid adding in further studies to the outcome map at this stage given the consensus process for the COS has recently completed. We believe that the Delphi offered appropriate opportunities to add in any outcomes of critical importance should any have been missed during the literature reviews of primary qualitative research.
8	Unfortunately, the literature search is restricted until May 2016, 4,5 years ago. Quite some interesting data on this subject came out more recently and would be of value to include in the future COS.	Whilst the literature review was conducted 4.5 years ago the COS project has been developing. The review work (both quantitative and qualitative), primary qualitative studies, and PROM coding took considerable time. As per the response above, given Delphi participants had the opportunity to add further outcomes during rounds of rating we believe that should any outcome of critical importance had been missed during the extensive list development work that it would have been added in during the Delphi. Therefore, the search being within the past 5 years is less of a concern as it might be for other projects e.g. meta-analysis of intervention effectiveness.
9	While the methodology section describes the qualitative research broadly (line204-226) and triggers the reader on the interviews, the result section is quite briefly on that subject while it is probably of great importance for a well-constructed COS.	The reviewer is correct that ensuring stakeholder (which included patients) input into COS development is of great importance, and we have ensured the perspective of gallstone patients are represented in the COS development through the interviews, focus groups, and analysis of clinical consultations. However, we have not conducted an in-depth analysis of the primary qualitative data to inform this development but rather a deductive content analysis that focused on outcomes of importance (as per methods section 237 – 244). The purpose of this paper is to produce an outcome map that characterises the outcomes generated across each of the evidence sources. This is also the case for other published outcome mapping processes during COS development.
10	Appendix 2 only show some patient characteristics, and the result section does not clearly state what is of most importance to most patients with gallstone disease.	Appendix 2 describes the demographics of the participants involved in the primary qualitative research: interviews, focus groups; and audio-consultations. As per previous comment, the purpose of this manuscript is to identify and categorise all of the outcomes with potential importance to a range of stakeholders. It was not to

		determine which outcomes were of most importance to any particular group. The final COS consensus process will establish which outcomes were most important to which groups and the community as a whole.
Reviewer #2:		Response
11	Thank you for the opportunity to review this paper. I think it is very well designed, conducted, and written. I think the methods were particularly well done and well described. I have only a couple of comments:	Thank you for this feedback.
12	Referring to studies used to extract endpoints: - Did any study describe the rationale for the selection of the reported outcomes? It is possible that some trialists have already deliberately dealt with the selection of the outcomes and this would be a relevant information here.	We did not collect data on rationale for selected outcomes. It is possible that the trials included in our literature review had considered selection of outcomes with regard to importance to stakeholders. However, we believe this is unlikely given the findings reported in Table 5 that demonstrate several of the outcomes reported in the quantitative evidence are not identified as important in the qualitative evidence (any source). With the converse also being true, that some outcomes identified by patients in the qualitative evidence are not reported by trials to date. However, we acknowledge that this could be a limitation of the work and have included text to highlight this on page 14 line 455-456.
13	- Did you consider the 'outcome reporting bias' when extracting the outcomes? Possibly, for the discussion and your intended process it could be interesting to know which outcomes were measured but not reported. The list of non-reported outcomes could be discussed in light of methodological and content-related aspects within the whole COS team	Thank you for highlighting this interesting issue. In this study we collated all the outcomes 'reported' in the papers, we did not record those listed as intended to be measured. We agree that an understanding of the non-reported outcomes would be relevant. We have undertaken this exercise in earlier papers that informed COS develop (Blencowe Ann Surgery 2013, Whistance Colorectal Disease 2013). We found that in surgical studies of oesophagectomy and colorectal cancer resection papers frequently did not report all the outcomes intended to be measured (50% at least or more did not do that). We believe that any outcomes of critical importance that have been missed through outcome reporting bias have other points at which they could be introduced to the COS. First through the inclusion of outcomes reported in the qualitative evidence (primary and secondary) and also through the ability

		of participants in the Delphi survey to introduce new outcomes for scoring in subsequent rounds (manuscript in preparation). As such, we don't believe it has negatively impacted on the COS development but would be important to consider in future studies. Therefore, we have updated the discussion of the paper to highlight this point and explain how in this specific study it was not done, in others it has been done and its something for future COS developers to consider. See page 14 line 450-454.
14	Referring to existing PROMS: - Did you analyze the development of these tools (cf. inauguration article)? According to COSMINs content validity tool, an excellent PROMs used patients' feedback to evaluate e.g. the relevance of items which are intended to include/which were included in a PROM. It could be a valuable resource to know whether or not patients were already involved in the development and how the relevance was rated.	We did extract whether the PROMS included in the content analysis had involved patients in their development. None of the included PROMS reported whether patients had been involved. We have now included a sentence in the results to highlight this (see page 11 line 350-351) in addition to the text already included in the discussion section (page 12 line 407-109).
15	Referring definition of outcomes: - On page 6 you describe that "any definition of outcomes" was extracted. I am confused because I don't understand how you harmonized the huge mess of reported outcome definitions. Please describe your approach in more detail.	The reported outcome definitions from included studies were used to inform the de-duplication of items from the long list. Text to make this clearer has now been included on page 5 line 158-160.
16	Others: - Page 9: "Participants were encouraged to consider what aspects of their disease or treatment impacted them most, both in terms of physical and social functioning and what improvements they would wish to see in terms of outcomes". Why were psychological aspects not addressed?	This was an error. We have now amended the text to read psychological instead of social. See page 8 line 239.
17	- Page 12: "This resulted in several outcomes being dropped from the long list as deemed not eligible as clinical endpoint outcomes for use in trials of this type (e.g. system and process outcomes such as duration of surgery which might be important in earlier phase trials)." Why did you put so much effort into the extraction of outcomes instead of using strict inclusion criteria from the beginning of your review?	We chose to keep the outcome extraction process broad so as to include all potentially relevant outcomes and publish a source of all outcomes relevant for gallstone disease, whether patients, clinical, or service level (Table 2). As this COS was also developed alongside a trial of surgery versus conservative management for gallstone disease we used it as an approach to ensure we weren't missing an outcome of importance in our trial.